# Is Narcissism Associated with Heavy Work Investment? The Moderating Role of Workload in the Relationship between Narcissism, Workaholism, and Work Engagement

**DOI:** 10.3390/ijerph17134750

**Published:** 2020-07-01

**Authors:** Alessandra Falco, Damiano Girardi, Annamaria Di Sipio, Vincenzo Calvo, Cristina Marogna, Raphael Snir

**Affiliations:** 1FISPPA Section of Applied Psychology, University of Padova, 35131 Padova, Italy; damiano.girardi@unipd.it (D.G.); annamaria.disipio@unipd.it (A.D.S.); vincenzo.calvo@unipd.it (V.C.); cristina.marogna@unipd.it (C.M.); 2School of Management and Economics, The Academic College of Tel Aviv-Yaffo, P.O.B 8401, Tel Aviv-Yaffo 6818211, Israel; rsnir@mta.ac.il

**Keywords:** narcissism, workaholism, work engagement, workload, moderation

## Abstract

This study aimed to investigate the association between narcissism and two forms of heavy work investment, namely, workaholism and work engagement. More specifically, it was hypothesized that narcissism is positively associated with both workaholism and work engagement, with workload moderating these relationships, which are expected to be stronger when the workload is high. Overall, 217 workers completed a self-report questionnaire, and the hypothesized relationships were tested using moderated multiple regression. Results partially supported our predictions. Narcissism was positively associated with workaholism and its dimensions of working excessively and working compulsively only in individuals facing a high workload. Furthermore, narcissism was positively associated with work engagement and its dimensions of vigor and dedication (but no absorption) in employees with average levels of workload. Finally, the workload exacerbated the relationship between narcissism and work engagement and its dimensions so that these associations were stronger when the workload was high. Overall, our study suggested that in a work environment characterized by moderate levels of demand, individuals with strong narcissistic components might inherently feel energetic and dedicated (i.e., engaged) at work. Differently, in a demanding work environment, workers with high narcissism might experience higher work engagement, but they could also be at risk of workaholism.

## 1. Introduction

Several scholars have recently argued that occidental culture has become more and more individualistic and self-focused in the past few decades [1]. It has been reasoned that narcissism may play a role of primary importance in such a phenomenon to the point that modern times may be facing a narcissism epidemic [2]. In support of this argument, it has been reported that among American college students, there has been a significant increase in narcissism since the 1970s [1,3]. However, this highly publicized empirical finding of the rise of narcissism in the United States is controversial and still largely under debate [4,5,6]. Moreover, there is no convincing evidence of a global increasing trend of narcissism in the world, outside the US, since it is not confirmed in other Western Countries, such as Australia and Canada [7]. Despite the contradiction of results on the current trend of narcissism [6], understanding deeply this construct and its influences on several aspects of everyday life remain an important issue of theoretical and empirical psychological research.

In the last years, the construct of narcissism conceptualized as a trait has been investigated in several organizational contexts. For instance, some studies have examined the complex interplay between narcissism and leadership. In particular, narcissism has been found to be linked to leadership positions, ethical leadership, and, with contrasting results, leader effectiveness [8,9,10]. Furthermore, previous studies have shown narcissism to be positively associated with perceived overqualification [11]. Similarly, some research has also been devoted to studying the influence of narcissism on work performance, focusing on three broad domains: task performance, organizational citizenship behaviors, and counterproductive work behaviors [10]. The results emerging from the extant studies are multifaceted; a complex picture emerges when considering the effect of narcissism on organizational variables, and more research is warranted.

Quite surprisingly, although with some exceptions [12,13], little attention has been devoted to the association between narcissism and heavy work investment (HWI), a concept that implies an elevated investment of both time and effort in one’s work [14]. However, one might expect individuals with strong narcissistic components to be also heavy work investors. Indeed, individuals with high narcissism may have a strong focus on being successful at work, an area of their life in which they can satisfy the need of power and admiration by demonstrating their abilities and superiority over others [13]. Hence, in this study, we hypothesized that narcissism conceptualized as a trait is positively associated with two specific types of HWI, namely, workaholism and work engagement, a “bad” and a “good” form of heavy work investment, respectively [15,16]. Furthermore, building on the biopsychosocial model [17,18,19], we also hypothesized that workload, a situational factor, may moderate the association between narcissism (i.e., an individual factor) and workaholism/work engagement, which is expected to be stronger when the workload is high.

### 1.1. Workaholism and Work Engagement

The aim of this study was to investigate the association between narcissism and two types of working hard, namely, workaholism and work engagement. On the one hand, workaholism may be defined as “the tendency to work excessively hard in a compulsive way” [20] (p. 204). Two central dimensions of the construct are identified, namely, working excessively (WE) and working compulsively (WC), and workaholism is characterized by high levels of both. Working excessively is the behavioral dimension of workaholism and reflects the fact that workaholics tend to dedicate an excessive amount of time to their work, and to work beyond what is expected of them to comply with economic or organizational requirements. Conversely, working compulsively is the cognitive dimension of workaholism, which indicates that workaholics are obsessed with work, and they insistently think about work even when they are involved in other activities [20]. On the other hand, work engagement may be conceived as a “positive, fulfilling, work-related state of mind that is characterized by vigor, dedication, and absorption” [21] (p. 74). Engaged employees are active, energetic, and work hard (vigor), they are enthusiastic and proud about their job (dedication), and they feel focused and happily engrossed in it (absorption) [15,16].

Although some researchers suggest workaholism to have some positive features [22,23], recent reviews [24] and empirical studies [25,26,27,28,29,30,31] have shown that workaholism is mostly associated with negative outcomes for both the individual and the organization, such as reduced job and life satisfaction, work-family conflict, physical and psychological symptoms, higher systolic blood pressure, cardiovascular risk, inflammatory response, sleep problems, reduced job performance, as well as sickness absences and presenteeism. On the contrary, work engagement is mostly associated with positive outcomes for the individual and the organization, such as physical and mental health, work ability, work-family enrichment, job and life satisfaction, organizational commitment, as well as task and contextual performance [29,32,33].

### 1.2. Heavy Work Investment and the Biopsychosocial Model

Snir and Harpaz [14] recently introduced the concept of heavy work investment, which is defined as the high investment of both time and effort (in terms of either physical or mental energy) in one’s work. In this perspective, workaholism and work engagement may be conceived as two different forms of dispositional heavy work investment [16]. This means that workaholism and work engagement primarily stems from predictors that are internal, rather than external, to the person. Accordingly, both workaholism and work engagement, by being dispositional HWI types, are especially relevant as outcome-variables of the individual phenomenon, such as narcissism. More specifically, workaholism is characterized by an (uncontrollable) inner drive to work hard, whereas work engagement is an expression of a (controllable) passion to work [20,34]. This is consistent with the idea that workaholism and work engagement are linked to obsessive and harmonious passion, respectively [35,36]. For these reasons, and given the distinct pattern of association with negative and positive outcomes described above, workaholism and work engagement are mainly regarded as a “bad” and a “good” form of heavy work investment, respectively [15,16].

Drawing on the theoretical perspective of Snir and Harpaz [14], recent research suggests different forms of HWI, including workaholism and work engagement, to be the combined result of both individual (internal) and situational (external) factors [15,24,37,38]. For example, Schaufeli [15] suggested that organizations might involuntarily encourage HWI in employees so that both personality traits and organizational aspects (e.g., organizational climate) might play a role in the onset of workaholism and work engagement. Moreover, building on the biopsychosocial model [17,18,19], a comprehensive theoretical framework according to which HWI can be conceived as the product of a complex interplay between biological, psychological, and social factors, we also hypothesized in this study the interaction between an individual (i.e., personality traits) and situational factors. In particular, we believed that workload, an aspect of the job that pushes employees to work hard, might facilitate heavy work investment in workers with high levels of narcissism, a personality characteristic that could predispose individuals to both workaholism and work engagement [12,13].

The biopsychosocial model [17] is the overarching framework, according to which we hypothesized this pattern of relationships. According to the biopsychosocial model, human behavior results from the interplay between biological, psychological, and social factors. In this perspective, McMillan and O’Driscoll [18] suggested that workaholism has a multifactorial genesis and may stem from complex interactions between personal dispositions (e.g., personality traits), cognitive and emotional processes, behaviors learned by the individuals, and the social systems in which they are embedded (e.g., the work context). Furthermore, Astakhova and Hogue [19] have recently proposed that different forms of HWI, including workaholism and work engagement (e.g., die-hard and activist sub-types of workaholic HWI, according to their typology), may result from the joint interplay of biological, psychological, and social factors, which include organizational and family factors. Hence, in line with the biopsychosocial model, personality dispositions, such as narcissism (and family experiences as well), may predispose individuals to heavy work investment, which is triggered by work-related factors, such as workload, and then maintained by cognitive, emotional, or behavioral processes.

### 1.3. Narcissism and Heavy Work Investment

The construct of narcissism is currently one of the most relevant and discussed in psychology. Freud’s [39] first use of the term “narcissism” in 1910 refers to individuals that “take themselves as sexual objects” (1905d, the note added in 1909, p. 460). In the psychodynamic field, Kohut and Kernberg made a central involvement in understanding the structure of the narcissistic personality. Otto Kernberg [40] agreed that the pathology focusses on the difficulty in the regulation of self-esteem and the persistence of a grandiose self but did not believe that this is the reactivation of a normal but pathological phase of childhood development. Whereas Kohut [41] sustained the existence of an “archaic grandiose self”, Kernberg spoke of “pathological grandiose self”, which must not be favored or allowed to grow but must be interpreted as a narcissistic character defense. Kohut [41] also showed the connection between borderline and narcissistic disorders with respect to the lack of attunements in the primary environment. This, during the psychic development, hinders the formation of the structures of internal self-regulation and determines the sense of stability and cohesiveness of the individual. For a long time, it was the psychoanalytic movement that dealt with narcissism, and only in 1980, with the DSM-III [42], the narcissistic personality officially became part of psychiatric diagnostics, perhaps because it was increasing in the population. Narcissistic personality disorder (NPD) was then maintained in DSM V [43], and for the first time, the description includes features of the type known as vulnerable or covert narcissism in addition to the already known overt narcissism.

In this study, we conceived narcissism “as an important complex of personality traits and processes that involve a grandiose yet fragile sense of self and entitlement as well as a preoccupation with success and demands for admiration” [44] (p. 440). Previous research has suggested that narcissism may play a role in the onset of workaholism and work engagement [12,13]. Possible mechanisms that explain the association between narcissism and the two forms of HWI could involve work motivation and cognitive processes. In this regard, previous research has shown that the motivational correlates of workaholism and work engagement are, in large part, different (although not necessarily completely distinct) [45,46,47]. For example, drawing on the self-determination theory (SDT) [48], van Beek et al. [46] found that workaholic employees were mostly driven by controlled motivation, including external and introjected regulation, whereas engaged workers were mostly driven by autonomous motivation, in terms of identified and intrinsic regulation. This implies that workaholics mainly perform their work for its instrumental value. More specifically, they work extremely hard to attain social rewards or to avoid punishment, such as social recognition or disapproval by others in the workplace, respectively (i.e., external regulation). Workaholics also work hard to meet internalized external standards of self-worth and social approval, in order to enhance feelings of worth and self-esteem as well as to avoid negative emotions related to failure (i.e., introjected regulation). Conversely, engaged workers find their work inherently satisfying and enjoyable, and they perform it for its own sake (i.e., intrinsic motivation). Furthermore, they identify themselves with the value of their work and recognize its importance, in terms, for example, of advancement in a career (i.e., identified regulation).

Interestingly, previous research has suggested that narcissism, which encompasses a complex set of personality characteristics, is positively associated with both autonomous and controlled motivation (for a review see Sedikides et al.) [49]. This implies that individuals with strong narcissistic components may perform their work activities for different reasons (e.g., because they enjoy their work and because of its instrumental importance for personal goals). Therefore, in the light of biopsychosocial model and given the above-mentioned link between different regulatory styles and heavy work investment, it seems reasonable to consider narcissism as a personal factor that may predispose individuals to invest a lot of time and effort to their work, in terms of both workaholism and work engagement. On the one hand, we expect narcissism to be positively associated with workaholism, a “bad” form of heavy work investment [15]. Individuals with strong narcissistic components are characterized by a chronic need to attain validation for their overly positive yet fragile self-image [50,51]. Hence, narcissists may work extremely hard to acquire external rewards and appreciation from others (e.g., social recognition, from colleagues or supervisor), as well as to avoid disapproval (i.e., external regulation). Moreover, individuals with strong narcissistic components may find themselves in a persistent state of ego-involvement, in which their self-esteem is contingent on their performance [51]. These individuals may feel pushed to work exceedingly hard to enhance feelings of self-worth, as well as to avoid negative emotions related to failure, such as anxiety and guilt (i.e., introjected regulation). For these reasons, individuals with strong narcissistic components may be predisposed to invest an exceedingly high amount of time in their work activities (i.e., working excessively) and to persistently think about their work, being obsessed with it (i.e., working compulsively). Stated differently, narcissism may be positively associated with “an obsessive, irresistible inner drive to work excessively hard” (i.e., workaholism) [20] (p. 219).

On the other hand, narcissism is expected to be positively associated with work engagement, a “good” form of heavy work investment [15]. Individuals with strong narcissistic components are characterized by the need for power and admiration, which may result in a keen focus on succeeding at work, an aspect of life in which they can overtly demonstrate their abilities [13]. They are also competitive and strive to establish their superiority over others [52,53]. Hence, narcissists may approach and enjoy their work activities, in which they can satisfy their need for power and admiration as well as their strong desire to demonstrate superiority over others, such as colleagues (i.e., intrinsic motivation) [51]. Furthermore, individuals with strong narcissistic components may be happily involved in their work because they recognize its importance as a means to achieve, success, advancement in career, and status (i.e., identified regulation). Overall, it is conceivable that individuals with strong narcissistic components, who may experience autonomous motivation to some extent [49], can be energetic and mentally resilient at work (i.e., vigor), enthusiastic and proud about their job (i.e., dedication), as well as focused and happily engrossed in it (i.e., absorption). All in all, narcissism may be positively associated with work engagement.

Empirical research, although still scarce, supports the idea of an association between narcissism and both workaholism and work engagement. Clark et al. [13] showed narcissism to be associated with overall workaholism and its dimensions of impatience and compulsion to work (but not polychronic control). In a study by Andreassen et al. [12], narcissism was found to be positively associated with work engagement and enjoyment of work but not associated with the drive. Overall, based on the arguments previously described and the results of previous research, we hypothesized that narcissism is positively associated with workaholism and its dimensions, that is, working excessively and working compulsively.

**Hypothesis** **1a** **(H1a):***Narcissism is positively associated with workaholism*;

**Hypothesis** **1b** **(H1b):***Narcissism is positively associated with working excessively*;

**Hypothesis** **1c** **(H1c):***Narcissism is positively associated with working compulsively*.

Furthermore, we also hypothesized that narcissism is positively associated with work engagement and its dimensions, that is, vigor, dedication, and absorption.

**Hypothesis** **2a** **(H2a):***Narcissism is positively associated with work engagement*;

**Hypothesis** **2b** **(H2b):***Narcissism is positively associated with vigor*;

**Hypothesis** **2c** **(H2c):***Narcissism is positively associated with dedication*;

**Hypothesis** **2d** **(H2d):***Narcissism is positively associated with absorption*.

### 1.4. The Moderating Role of Workload

Based on the biopsychosocial model, we expected workload to moderate the association between narcissism and the two forms of heavy work investment, namely, workaholism and work engagement. There are several mechanisms that could underlie this interaction between individual and situational factors. First, with respect to workaholism, the workload may signal what the norms are in an organization or in a work team (i.e., to work extremely hard) [54]. Hence, the high workload may encourage workaholism in employees with strong narcissistic components, who may feel pushed to work hard to achieve social recognition and rewards, as well as to attain ego enhancement, by meeting these extremely demanding external standards. Accordingly, it is conceivable that in a demanding work environment (e.g., elevated expectations in terms of productivity), individuals with strong narcissistic components may have the tendency to work exceedingly hard (i.e., working excessively) and to feel obsessed with their work (i.e., working compulsively), given their chronic need to attain validation for their overly positive yet fragile self-image and their persistent state of ego-involvement, in which self-esteem is contingent on performance outcomes [51].

Second, it is possible that individuals with strong narcissistic components experience work engagement, especially when the workload is high. This is consistent with the idea that intrinsic motivation, a possible motivational correlate of work engagement, arises when the goals of an individual match the goals supported by the environment [55]. Hence, employees high in narcissism may particularly approach and enjoy work situations in which workload is high because a demanding work environment, characterized by elevated expectations in terms of productivity and performance, may offer workers with high narcissism the opportunity to satisfy their strong desire of demonstrating superiority over others (e.g., by being more productive than colleagues) [51,56]. This matching between the goals of the individual and that supported by the work environment, in turn, may result in higher levels of intrinsic motivation and work engagement. Accordingly, individuals with strong narcissistic components may feel energized (i.e., vigor) and enthusiastic (i.e., dedication) at work, as well as happily engrossed in it (i.e., absorption), especially when the workload is high.

Overall, based on these arguments and given the assumptions of the biopsychosocial model, we hypothesized that workload moderates the association between narcissism and workaholism, as well as its dimensions (i.e., WE and WC), with this association being stronger for workers with a higher workload.

**Hypothesis** **3a** **(H3a):***Workload moderates the association between narcissism and workaholism, which is expected to be stronger when the workload is high*;

**Hypothesis** **3b** **(H3b):***Workload moderates the association between narcissism and working excessively, which is expected to be stronger when the workload is high*;

**Hypothesis** **3c** **(H3c):***Workload moderates the association between narcissism and working compulsively, which is expected to be stronger when the workload is high*.

We also hypothesized that workload moderates the association between narcissism and work engagement, as well as its dimensions (i.e., vigor, dedication, and absorption), with this association being stronger for workers with a higher workload.

**Hypothesis** **4a** **(H4a):***Workload moderates the association between narcissism and work engagement, which is expected to be stronger when the workload is high*;

**Hypothesis** **4b** **(H4b):***Workload moderates the association between narcissism and vigor, which is expected to be stronger when the workload is high*;

**Hypothesis** **4c** **(H4c):***Workload moderates the association between narcissism and dedication, which is expected to be stronger when the workload is high*;

**Hypothesis** **4d** **(H4d):***Workload moderates the association between narcissism and absorption, which is expected to be stronger when the workload is high*.

To the best of our knowledge, no prior empirical research has explored the above-mentioned possible pattern of relationships. Although some previous studies have investigated the association between narcissism (along with other personality traits) and workaholism [13], as well as the association between narcissism and workaholism/work engagement [12], we believed that the value added by this study was that we examined the interplay between two personal and situational factors, namely, narcissism and workload, in explaining the two forms of heavy work investment.

## 2. Materials and Methods

### 2.1. Procedure and Participants

The present study was conducted on a sample of workers from different organizations in Italy. Participants were approached by trained research assistants and were invited to complete an anonymous questionnaire (paper-and-pencil) about their work experience. Overall, two hundred and thirty-seven questionnaires were completed. However, twenty participants had missing values in at least one of the variables considered in the study. Data were missing completely at random (Little’s missing completely at random test, χ^2^ = 992.37, *df* = 1013, *p* = 0.67), and cases with missing values on any of the measures considered in the study were removed from the dataset (i.e., listwise deletion). Accordingly, the final sample comprised of 217 participants. This sample consisted of 121 women (55.7%) and 95 men (43.8%; one missing value, 0.5%) with a mean age of 45 (*SD* = 12). Concerning the type of job (i.e., intellectual or manual), 57.1% were freelancers, managers, or white-collar workers, 20.3% were blue-collar workers, whereas 22.6% were from other professions. Regarding the type of contract, 190 workers (87.6%) had a permanent contract, whereas 25 (11.5%) had a temporary contract (2 missing values, 0.9%). With respect to work experience, 38.7% had been with their current company for more than 20 years and 30.9% for less than 10 years (4 missing values, 1.8%).

### 2.2. Measures

The following self-report measures were administered:

Narcissism was assessed using the Italian adaptation of a short form of the narcissistic personality inventory (NPI) [57], namely, the NPI-16 [44]. The instrument in its original form included sixteen narcissism-consistent responses coded as 1 (e.g., “I know that I am good because everybody keeps telling me so”) and sixteen narcissism-inconsistent responses coded as 0 (e.g., “When people compliment me, I sometimes get embarrassed”). Items were summed, and narcissism was measured as the total composite scale score.

Workaholism was measured by using the Dutch work addiction scale (DUWAS) [20] in the Italian adaptation [58,59]. The scale was composed of ten items, designed to detect the two dimensions of WE (six items; e.g., “I seem to be in a hurry and racing against the clock”) and WC (four items; e.g., “I feel that there’s something inside me that drives me to work hard”). The six-point response scale ranged from 1 (strongly disagree) to 6 (strongly agree). Since workaholism reflects the tendency to work excessively hard in a compulsive way, an overall workaholism score was used, in addition to the single scores of WE and WC.

Work engagement was determined using the Italian adaptation [60] of the short version of the Utrecht work engagement scale [21], that is, the UWES-9 [61]. The scale was composed of nine items, designed to detect the three dimensions of vigor (three items; e.g., “At my job, I feel strong and vigorous”), dedication (three items; e.g., “I am enthusiastic about my job”), and absorption (three items; e.g., “I am immersed in my job”). In the present study, the six-point response scale ranged from 1 (never) to 6 (always). According to Schaufeli et al. [61], the overall work engagement score was used for this study, in addition to the single scores of vigor, dedication, and absorption.

The workload was measured by using a scale taken from the Q_u_-Bo test, an instrument standardized for the Italian context [62]. The scale was composed of three items (e.g., “Your job requires you to work very fast”), and the six-point response scale ranged from 1 (strongly disagree) to 6 (strongly agree).

### 2.3. Ethical Aspects

This research was in accordance with the national law about data treatment, which is rigorously followed by the University of Padova (Italy). Additional ethical approval was not required, as there were no medical treatments or other procedures that could cause psychological or social discomfort to participants, who were healthy adult individuals who took part in the study anonymously. This research was conducted in line with the Helsinki Declaration as well as the Italian data protection regulation (Legislative Decree No. 196/2003). Participants decided to take part in the research on a voluntary basis and did not receive any reward. Data collection and analysis were anonymous. A cover letter attached to the questionnaire provided information about the purpose of the study, anonymity, voluntary participation, as well as data treatment. All participants were made aware that they could withdraw from the research at any time, and they gave their written informed consent before the administration of the questionnaire.

### 2.4. Data Analysis

Prior to estimating the regression models aimed at hypothesis testing, several confirmatory factor analyses (CFA) were carried out to evaluate the psychometric properties of the self-report questionnaires administered in the study. Two different approaches were adopted for workaholism/work engagement on the one hand, and narcissism on the other. With respect to workaholism and work engagement (i.e., DUWAS and UWES-9), the maximum likelihood method with robust standard errors and scaled test statistic (i.e., robust maximum likelihood, MLR) was the estimator. Indeed, past research has shown that robust maximum likelihood has adequate performance when analyzing Likert scales with five or more categories [63]. Differently, to account for the dichotomous nature of the NPI items (i.e., narcissism), CFAs were conducted using the mean- and variance-adjusted weighted least squares estimator (i.e., WLSMV) with tetrachoric correlations. In order to evaluate model fit for workaholism/work engagement, a scaled test statistic (χ^2^) was used, together with additional fit indices, namely, the root mean square error of approximation (RMSEA), the comparative fit index (CFI), and the standardized root mean square residual (SRMR). A model shows a good fit to data if the χ^2^ is non-significant. Additionally, values close to or smaller than 0.08 for RMSEA and SRMR, as well as values close to or greater than 0.90 for CFI, indicate an acceptable fit [64]. A similar approach was adopted for narcissism; the only difference being that the weighted root mean square residual (WRMR) was considered in place of SRMR. Indeed, WRMR showed adequate performance, especially when dichotomous data were analyzed [65]. For WRMR, values close to or smaller than 1 indicate an acceptable fit [65].

Then, the hypothesized relationships were tested using moderated multiple regression analyses following the procedure outlined by Aiken and West [66]. Overall, seven different models were estimated. In model 1 (M1), workaholism was the dependent variable, whereas narcissism and workload were the independent and the moderating variables, respectively. Model 2 (M2) and model 3 (M3) were similar to M1, except that working excessively (M2) or working compulsively (M3) were the dependent variables, respectively. In the same way, work engagement was the dependent variable in model 4 (M4), and narcissism and workload were the independent and the moderating variables, respectively. Model 5 (M5), model 6 (M6), and model 7 (M7) were similar to M4, except that vigor (M5), dedication (M6), or absorption (M7) were the dependent variables, respectively. In all the models tested (M1–M7), the scores of both narcissism and workload were centered, and then the cross-product of centered variables was computed. The models were estimated, both including and omitting the respective interaction term, to assess the additional variance explained by each of them. If a significant interaction was found, then a simple slope analysis was conducted to determine whether narcissism was associated with the different forms of HWI (i.e., workaholism, work engagement, as well as their dimensions) at high (+1*SD*) and low (−1*SD*) levels of workload [66]. Finally, to interpret the nature of the moderating effect, significant interactions were presented graphically, following the procedure outlined by Aiken and West [66]. Because previous studies have suggested that demographic characteristics may be associated with narcissism and HWI, all the regression models were estimated, both controlling for and not controlling for gender and age. Given that the results were very similar, only the more parsimonious models (i.e., without controlling for the effect of gender and age) were presented. Statistical analyses were carried out using the software R version 3.6.2 [67], and, more specifically, CFAs were carried out using the lavaan package version 0.6-5 [68] for R software.

## 3. Results

### 3.1. Confirmatory Factor Analysis

First, a CFA was carried out to investigate the psychometric properties of the DUWAS. In this model, working excessively and working compulsively were measured by the ten items of the DUWAS (six for WE and four for WC, respectively). The fit indices showed an acceptable fit to data: χ^2^(34) = 53.47, *p* = 0.02; RMSEA = 0.05, CFI = 0.97, SRMR = 0.04. Then, the psychometric properties of the UWES-9 were investigated. In this model, the three dimensions of vigor, dedication, and absorption were measured with three items each. The fit indices showed an acceptable fit to data: χ^2^(24) = 37.91, *p* = 0.04; RMSEA = 0.05, CFI = 0.98, SRMR = 0.04. Finally, a CFA was carried out to investigate whether the two forms of HWI could be distinguished empirically. In this model, working excessively and working compulsively were the two indicators of workaholism, whereas vigor, dedication, and absorption were the indicators of work engagement. The fit indices showed a poor fit to data: χ^2^(4) = 50.04, *p* < 0.001; RMSEA = 0.23, CFI = 0.89, SRMR = 0.07. A closer inspection of the modification indices revealed a substantial cross-loading for absorption on workaholism. Given that previous research showed similar results [69], the model was estimated again, including this cross-loading. The fit indices showed now an acceptable fit to data: χ^2^(3) = 1.57, *p* = 0.67; RMSEA = 0, CFI = 1, SRMR = 0.01. Then, a CFA was carried out to investigate the psychometric properties of the short version of the NPI. The fit indices showed an acceptable fit to data, except for WRMR: χ^2^(104) = 180.10, *p* < 0.001; RMSEA = 0.06, CFI = 0.93, WRMR = 1.09. A closer inspection of the tetrachoric correlation matrix and the modification indices revealed that three items behaved poorly. These items were removed, and a new CFA was performed. Fit indices showed now an acceptable fit to data: χ^2^(65) = 100.16, *p* < 0.01; RMSEA = 0.05, CFI = 0.93, WRMR = 0.95. Therefore, the revised short form of the NPI (13 items) was used in the subsequent analyses. Overall, the instruments administered in the study showed good psychometric properties.

### 3.2. Hypothesis Testing

Descriptive statistics, correlations between study variables, as well as Cronbach’s alphas, are reported in Table 1. Narcissism was positively associated with workaholism (*r*_215_ = 0.15, *p* < 0.05) and working excessively (*r*_215_ = 0.14, *p* < 0.05), but not working compulsively. Furthermore, narcissism was also positively associated with work engagement (*r*_215_ = 0.17, *p* < 0.05), vigor (*r*_215_ = 0.19, *p* < 0.01), and dedication (*r*_215_ = 0.15, *p* < 0.05), but not absorption. Overall, the size of these correlations was small [66].

Results of the regression analyses for workaholism and its dimensions of WE and WC (M1–M3) are presented in Table 2. In these models, narcissism was not associated either with workaholism or working excessively/compulsively after controlling for the effect of both workload and the interaction term. Hypothesis 1 (H1a, H1b, H1c) was not supported.

Results of the regression analyses for work engagement (M4) and its dimensions of vigor (M5), dedication (M6), and absorption (M7) are shown in Table 3. In these models, after controlling for the effect of both workload and the interaction term, narcissism was positively associated with work engagement (*b* = 0.06, *p* < 0.05), vigor (*b* = 0.08, *p* < 0.01), and dedication (*b* = 0.07, *p* < 0.05), but not absorption. Hence, hypotheses 2a, 2b, and 2c were supported, whereas hypothesis 2d was not.

With respect to H3, the interaction between narcissism and workload was significant in M1 and accounted for an additional 4.3% of the variance in workaholism, *F*_change_(1, 213) = 12.24, *p* < 0.001. An analogous pattern of results was found for both WE and WC. In M2, the interaction between narcissism and workload was significant and accounted for an additional 2.2% of the variance in working excessively, *F*_change_(1, 213) = 6.17, *p* < 0.05. Similarly, the interaction between narcissism and the workload was significant also in M3 and accounted for an additional 5.3% of the variance in working compulsively, *F*_change_(1, 213) = 13.03, *p* < 0.001.

The simple slope analysis showed that the relationship between narcissism and workaholism (M1) was positive and significant when the workload was high (*b* = 0.12, *p* < 0.001) but nonsignificant when the workload was low. Similar results also occurred for WE (M2) and WC (M3). Indeed, the simple slope analysis showed that, when the workload was high, narcissism was positively associated with both working excessively (*b* = 0.11, *p* < 0.01) and working compulsively (*b* = 0.14, *p* < 0.001). However, when the workload was low, narcissism was not associated either with working excessively or working compulsively. To further interpret the nature of the moderating effect, the interactions were presented graphically (Figure 1). The association between narcissism and workaholism was stronger for individuals with high levels of workload (Figure 1a). The same pattern also occurred for working excessively (Figure 1b) and working compulsively (Figure 1c). Overall, hypothesis 3 (H3a, H3b, and H3c) was supported.

With respect to H4, the interaction between narcissism and workload was significant in M4 and accounted for an additional 3.4% of the variance in work engagement, *F*_change_(1, 213) = 7.73, *p* < 0.01. An analogous pattern of results was found for vigor, dedication, and absorption. In M5, the interaction between narcissism and workload was significant and accounted for an additional 4.5% of the variance in vigor, *F*_change_(1, 213) = 10.40, *p* < 0.01. Similarly, the interaction between narcissism and workload accounted for an additional 2.4% of the variance in dedication (M6), *F*_change_(1, 213) = 5.32, *p* < 0.05, and the same also occurred in M7, in which the interaction term explained an additional 1.9% of the variance in absorption, *F*_change_(1, 213) = 4.20, *p* < 0.05.

The simple slope analysis showed that the relationship between narcissism and work engagement (M4) was positive and significant when the workload was high (*b* = 0.14, *p* < 0.001) but nonsignificant when the workload was low. Similar results also occurred for vigor (M5), dedication (M6), and absorption (M7). Indeed, the simple slope analysis showed that, when the workload was high, narcissism was positively associated with vigor (*b* = 0.17, *p* < 0.001), dedication (*b* = 0.15, *p* < 0.01) as well as absorption (*b* = 0.11, *p* < 0.01). However, when the workload was low, the associations between narcissism on the one hand and vigor, dedication, and absorption on the other were not significant. Finally, to further interpret the nature of the moderating effect, the interactions were presented graphically (Figure 2). The association between narcissism and work engagement was stronger for individuals with high levels of workload (Figure 2a). The same pattern also occurred for vigor (Figure 2b), dedication (Figure 2c), and absorption (Figure 2d). Overall, hypothesis 4 (H4a, H4b, H4c, and H4d) was supported.

## 4. Discussion

This study investigated the association between narcissism, workload, and two types of working hard, namely, workaholism and work engagement. Building on the heavy work investment [14] and the biopsychosocial models [17,18,19], it was hypothesized that narcissism is positively associated with two types of HWI, namely, workaholism and work engagement, with workload moderating these relationships, which are expected to be stronger when the workload is high. Results partially supported our predictions. In fact, on the one hand, narcissism was positively associated with workaholism and its dimension of WE and WC only in individuals facing a high workload. On the other hand, narcissism was positively associated with work engagement and its dimension of vigor and dedication (but not absorption) in employees with average levels of workload. Furthermore, workload moderated the relationship between narcissism and work engagement as well as its dimensions (including absorption), so that these associations were stronger when the workload was high. Taken together, our results suggested that narcissism per se might give rise to work engagement but not to workaholism. However, narcissism might be a risk factor for workaholism in the presence of a demanding work environment (i.e., high workload). Furthermore, a demanding work environment might also exacerbate the association between narcissism and work engagement.

A possible explanation for this pattern of association is that in a work environment characterized by moderate levels of demand (i.e., for average levels of workload), individuals with strong narcissistic components may inherently approach and enjoy their work, in which they can satisfy their need of power and admiration as well as their strong desire of demonstrating superiority over others (e.g., colleagues) [13]. In other words, in these situations, narcissism may be associated with work engagement, not workaholism. This is consistent with a previous study by Andreassen et al. [12], who found narcissism to be positively associated with enjoyment of work and work engagement, but unrelated to drive, a core characteristic of workaholism that is especially associated with negative outcomes for the individual (e.g., emotional exhaustion) [47].

Differently, in a demanding work environment (i.e., when the workload is high), narcissism may lead to the onset of workaholism. In these situations, individuals with strong narcissistic components, who are characterized by a chronic need to attain validation for their overly positive yet fragile self-image [51], may feel pushed to work exceedingly hard to achieve social recognition and rewards, as well as to attain ego enhancement, by meeting the extremely demanding external standards (e.g., elevated expectations in terms of productivity and performance). Put differently, when the workload is high, narcissism may play a role in the development of an obsessive inner drive to work hard. Interestingly, previous research on the relationship between perfectionism and workaholism has shown similar results. For example, Falco et al. [70] found in a longitudinal study that self-oriented perfectionism predicted an increase in workaholism over time only in workers facing a high workload. Furthermore, in a demanding work environment, individuals with strong narcissistic components may also experience higher work engagement. Indeed, intrinsic motivation, a possible motivational correlate of work engagement, arises when the goals of an individual match the goals supported by the environment [55]. Thus, a work environment characterized by high expectations in terms of performance and productivity may provide workers with high narcissism the opportunity to satisfy their strong desire of demonstrating superiority over others (e.g., by being more productive than colleagues), thus fostering intrinsic motivation and work engagement [51,56]. All in all, our results indicated that narcissism might be associated with both “bad” and “good” forms of HWI (e.g., workaholism and work engagement, respectively). This was consistent with the idea that narcissism (especially its grandiose aspects, which are predominantly measured by the NPI) [71] may be conceived as a double-edged sword, with both adaptive and maladaptive outcomes [72]. Furthermore, our study suggested that situational factors, such as workload, might influence how, and under what circumstances, narcissistic traits and processes could be channeled into either positive or negative forms of HWI.

These findings provided a valuable contribution to the existing literature on narcissism and HWI for several reasons. First, taken together, our results were consistent with the biopsychosocial model applied to HWI [14,18,19]. In this perspective, different forms of heavy work investment may stem from complex interactions between the individual (e.g., personality traits) and situational (e.g., organizational) factors. In line with the model’s prediction, our study showed that workaholism and work engagement might especially arise when individuals with strong narcissistic components, which might predispose individuals to HWI, faced high levels of workload, an aspect of the job that stimulate or compel employees to work hard. Nevertheless, it is quite reasonable that narcissism by itself will not lead to workaholism. After all, heavy investment in work does not serve self-admiration. However, a high workload might endanger one’s impression management (e.g., one’s supervisor would think that one was not coping properly with his/her workload and that his/her performance was poor), and thus trigger workaholism. This trigger in the situation of high workload is also relevant regarding work engagement, as both types of heavy work investors derive their self-esteem from their work. The distinction between workaholism and work engagement is that even average workload will trigger work engagement among narcissists since that engagement is based on intrinsic motivation (in other words, the average workload does not threaten them, it challenges and excites them). Interestingly, although previous research has investigated the association between personality traits and dispositions (including narcissism) on the one hand, and workaholism/work engagement on the other [12,13,15,73,74,75,76,77], this was the first study that examined the interaction between narcissism and workload in explaining the two forms of HWI.

Second, our study showed that workload exacerbated the positive association between narcissism on the one hand and workaholism as well as work engagement on the other. Put differently, in a demanding work environment, narcissism may be associated with both workaholism and work engagement. Although these findings may appear counterintuitive, it should be considered that workaholism and work engagement could be present simultaneously in the same individual [46,78]. For example, drawing on the concepts of workaholism and work engagement, van Beek et al. [46] distinguished between four profiles of workers. Those were workaholic employees (who are workaholics and not engaged), engaged employees (engaged and nonworkaholics), engaged workaholics (both engaged and workaholic), and nonworkaholic/non-engaged workers (nonworkaholics and not engaged). Interestingly, the authors found that engaged workaholics were driven by autonomous as well as controlled motivation. Moreover, although engaged workaholics dedicated the most time to their work, they also reported less burnout than workaholics employees (although more burnout compared to engaged employees). In light of these results, van Beek et al. [46] suggested that work engagement might actually buffer against the negative consequences of workaholism. Furthermore, Loscalzo and Giannini [78] proposed a similar conceptualization based on the HWI model. Hence, previous studies suggest that workaholism and work engagement are two relatively distinct phenomena that can co-occur in the same individual. Interestingly, past research has also shown that personality traits may influence the perception of the work environment [79] and that several job demands, such as time pressure, can be perceived as both opportunities to grow, achieve, and demonstrate one’s competences (i.e., challenging stressors), as well as barriers that unnecessarily hamper one’s progress toward goal attainment and rewards resulting from being evaluated as an effective performer (i.e., hindrance stressors) [80,81]. Therefore, in the light of previous research, it is reasonable to speculate that individuals with strong narcissistic components may perceive high workload both as an opportunity to fulfill their strong desire of demonstrating superiority over others as well as a high standard of performance that must be necessarily overcome, in order to satisfy the need for recognition and to avoid negative emotions related with failure. Put differently, narcissism, which encompasses a complex set of personality characteristics, may be associated with the appraisal of workload as an aspect of work that is both challenging and hindering. This may promote autonomous as well as controlled forms of work motivation (e.g., through satisfaction/frustration of basic psychological needs) [82,83], which may be mostly associated with work engagement and workaholism, respectively [45,46,47]. Hence, it is conceivable that, in a demanding work environment, individuals with strong narcissistic components may be both workaholic and engaged in their work; that is, they are engaged workaholics. Notably, engaged workaholics may experience lower levels of negative outcomes compared to disengaged workaholics [46], given that work engagement may buffer against the negative consequences typical of workaholism [27,84]. This is consistent with the idea that engaged workaholics have higher levels of resources (e.g., energies, conditions, and personal characteristics) than disengaged workaholics, and that individuals with more resources are better able to deal with the stressful situation than those with fewer resources [85]. Accordingly, Loscalzo and Giannini [78] suggested that disengaged workaholics, but not engaged workaholics, should be considered as the “real” or clinical workaholics. Hence, all in all, our study suggested that in a demanding work environment, narcissism might be associated with a less severe form of workaholism, characterized by high levels of both workaholism and work engagement.

Third, when examining the relationship between narcissism and each dimension of work engagement, a different picture emerged for absorption, compared to vigor and dedication. Indeed, narcissism was associated with absorption only when the workload was high, whereas the relationship between narcissism and vigor/dedication (as well as overall work engagement) was significant even for average levels of workload. A possible explanation is that employees with strong narcissistic components may inherently feel energetic and dedicated at work (i.e., vigor and dedication, the two core dimensions of work engagement) [86], an aspect of life in which they can demonstrate their abilities and satisfy their needs and desires, such as power, admiration, and superiority [13]. However, in a demanding work environment, characterized by elevated expectations in terms of productiveness and performance (i.e., when the workload is high), individuals with high narcissism may strive to satisfy their needs and desires by being more productive than others. Consequently, they may feel even more energetic and dedicated (e.g., given the match between their goals and those supported by the environment) [55], but they may also fully concentrate on their work, having difficulties in detaching themselves from it (i.e., absorption). Interestingly, our results regarding absorption were similar to those observed for workaholism, as well as its dimensions of WE/WC. Although unexpected, this finding was consistent with the idea of an overlap between workaholism and work engagement, with the latter having a possible dark side. Notably, in a recent meta-analysis, Di Stefano and Gaudiino [87] found absorption to be positively associated with both WE and WC. The authors suggested that absorption referred to the difficulty in disengaging from work and implied an intense investment of time in one’s work, two aspects that are also related to the cognitive and behavioral component of workaholism, respectively. Overall, in the light of previous studies, it is not surprising that absorption, similarly to workaholism, may arise when individuals with strong narcissistic components face high levels of workload (i.e., elevated expectations in terms of productivity and performance). Indeed, like workaholism, absorption implies being fully immersed in one’s work and having difficulties in detaching from it (although the underlying work motivation should be different) [69].

Some aspects of this study warrant further consideration. First, it should be considered that narcissism was determined using the narcissism personality inventory, an instrument that was originally designed to measure individual differences in narcissism in the nonclinical population [57]. Therefore, the NPI is not aimed at assessing pathological narcissism, and it may not completely address the maladaptive aspects of personality associated with narcissism [56,88]. This should be kept in mind when interpreting the results of the study. However, despite some criticism [89,90], the NPI is a valid and widely used instrument [91] that was also adopted in previous studies that investigated the association between narcissism and workaholism/work engagement [12,13]. Overall, further research is recommended to replicate and extend our findings.

Second, cultural dimensions might have affected the results of this study. Indeed, it has been proposed that culture may influence HWI, as well as its antecedents and outcomes [92,93,94]. More specifically, the dimension of collectivism/individualism may play a central role in this process [93]. According to Hofstede et al. [95] (p. 92), individualism refers to “societies in which the ties between individuals are loose: everyone is expected to look after him- or herself and his or her immediate family”. On the contrary, collectivism “pertains to societies in which people from birth onward are integrated into strong, cohesive in-groups, which throughout people’s lifetime continue to protect them in exchange for unquestioning loyalty”. Although work is considered to play a central role in people’s lives across cultures, it has been proposed that the motivation underlying working hard may be different in collectivistic versus individualistic societies [93]. According to Hu et al. [93], in individualistic cultures (e.g., Western Europe), workers emphasize personal goals and individual achievements, and they work hard because they are driven by the self-centered motivation to satisfy their needs for development and personal growth. Conversely, collectivistic cultures (e.g., countries of Eastern Asia, such as China and Japan) emphasize group ties that involve reciprocal obligations between members. In these societies, working hard is fueled by the group-centered motivation for social approval (e.g., to meet the expectations of one’s organization or work-team). Notably, this study was carried out in a sample of workers from the north of Italy, and Italy is considered an individualistic culture (especially its northern regions) [95]. Accordingly, this study might overestimate the role of narcissism, which could be a relevant antecedent of HWI, especially in individualistic cultures. Indeed, as pointed out by Ehrenberg [96] (p. 98), “the double movement of the decline in social obligation and the decline in collective solidarity is said to be shown in a new symptomatology characterised by an unprecedented disorganisation of the individual personality by the yardstick of narcissism. We are said to have entered an individualism that no longer involves personalisation but unbinding”. Furthermore, it is possible that a high workload may encourage work engagement, especially in individualistic societies, but workaholism in collectivistic societies. All in all, we believed that the present study should be replicated in different cultural contexts (e.g., in collectivistic societies).

Third, in this study, narcissism scores were not particularly high when compared to the scale range (0–13) and the results by Ames et al. [44]. Therefore, we could not rule out the possibility that the present findings did not generalize across different population groups, especially with respect to those with high levels of trait narcissism. Interestingly, the original study by Ames et al. [44] was carried out in the United States, a cultural context characterized by high levels of individualism [95], which could encourage narcissistic traits and processes. We believed that future research should address these questions. For example, participants with higher levels of narcissism could be enrolled, and the study could be replicated in cultural contexts that are characterized by higher levels of individualism (e.g., United States, Australia, United Kingdom).

Furthermore, the workload was conceived in the study as a situational variable. However, it should be noted that the relationship between workload and HWI is more complex, and different conceptualizations are possible. For example, according to the dynamic interactionist approach, heavy work investors may choose and stay in demanding jobs [23]. Furthermore, workaholics may create additional work for themselves, in an attempt to continue working (e.g., by not delegating their work) [20]. Finally, personality traits associated with workaholism (e.g., neuroticism, negative affectivity) may influence the perception of job demands, so that workaholics may perceive their jobs as having high levels of workload [79]. Taken together, these different conceptualizations suggest that future research is needed to further understand the complex association between workload and HWI.

Finally, gender differences and gender roles might play a role in the processes investigated in this study. For example, past research has shown that men have higher levels of narcissism than women [4]. On the contrary, women may perceive their work environment as more stressful [79], and they may also have a prolonged psychophysiological response to stress [97]. Furthermore, it has been proposed that, according to societal norms, men are expected to be more invested in work, while women are expected to be more invested in family (for a review, see Clark et al.) [25]. All in all, although the exploration of gender differences was beyond the scope of this study, we believed that future research should investigate specific mechanisms through which gender differences and gender roles might affect the genesis of different forms of HWI.

This study had some limitations. First, the focal constructs were determined using the same measurement method, that is, self-report questionnaires. Hence, the observed relationships could be affected by method bias [98]. However, as pointed out by Podsakoff et al. [98], one strategy to control for method bias is to minimize the scale properties shared by the measures of the predictor and the criterion variables (e.g., scale type, number of scale points). Notably, in this study, the instruments aimed at determining the independent (i.e., narcissism) and the dependent variables (i.e., workaholism and work engagement) adopted very different response formats (i.e., dichotomous items vs. Likert scales, respectively), which should reduce common method bias. However, although we believed that method bias should not be a major concern in this study, future research could adopt, for example, observer-ratings (e.g., spouse rating) or biomarkers as measures of workaholism and work engagement [58,99,100,101,102]. Moreover, the cross-sectional design of the study precluded conclusions about the direction of the observed relationships. Although the biopsychosocial model assumes that heavy work investment may result from the interplay between biological, psychological, and social factors [18,19], future studies should further investigate the longitudinal relationships between both individual and situational factors on the one hand and workaholism/work engagement on the other, given that these phenomena may influence and reinforce each other over time [25]. Finally, in this study, we focused on a specific aspect of the job, namely, workload. However, it seems plausible that other characteristics of the work environment (e.g., overwork climate) [15], as well as specific supervisor’s leadership styles (e.g., destructive and transformational leadership) [103], may affect the association between narcissism and heavy work investment.

Finally, this paper had several practical implications for practitioners. In this perspective, we believed that the most relevant finding of our study was that, in a demanding work environment, individuals with strong narcissistic components might be both workaholic and engaged in their work, that is, they might be engaged workaholics. Although Loscalzo and Giannini [78] considered only disengaged workaholics (i.e., those with high workaholism and low work engagement) as the “real” or clinical workaholics, interventions should be aimed at dealing with both narcissism and job demand at work, to prevent the onset of a more severe or clinical form of workaholism in individuals with strong narcissistic components. More specifically, when dealing with narcissism in organizational contexts, preventive measures are suggested to be much more effective than curative ones [104]. In general terms, interventions aiming at reducing the negative consequences of individuals’ narcissistic traits and characteristics should foster the development of an organization’s culture that stigmatizes self-serving and narcissistic behaviors and promotes calm collaboration among co-workers [104]. Interestingly, experimental laboratory investigations suggest that increasing the sense of connection to others in narcissistic individuals could be an important target of intervention. In particular, an augmented sense of connection to others seems to mitigate narcissists’ aggressive behaviors [105] and to promote relationship commitment [106]. Therefore, interventions aimed to develop a solid sense of ‘‘belonging together’’ [105,107] could be effective in preventing and/or mitigating the negative consequences of narcissism in the workplace. These kinds of interventions could be delivered in the form of group sessions. Additionally, interventions could help workers with strong narcissistic components to develop skills to cope with high workload (e.g., time management, goal-setting, and biofeedback), which may enhance control and reduce negative emotions at work.

Moreover, this study suggested that organizations should target narcissism, especially in managers. Indeed, narcissistic leaders, who necessarily face high job demands in their job, are particularly at risk of workaholism [108], and workaholic managers may negatively impact the organizations they work for. Accordingly, organizations should be encouraged to consider the role of narcissistic leaders in their relationship with subordinates. Individuals with strong narcissistic components present prototypical characteristics associated with leadership that could facilitate them in selection contexts, but they also have many negative interpersonal characteristics that can be destructive on daily functioning at work. Indeed, these individuals experience a need for power that may lead to an excessive focus on work, they have aggressive tendencies, and they are also critical of others and demand perfectionism [13,109,110]. Subordinates perceive this as a destructive and abusive supervision style that may be related to psychological distress, emotional exhaustion, and lower task performance. Therefore, organizations should propose interventions aimed at preventing a bad fit between narcissistic managers and vulnerable subordinates (e.g., creating teams with managers lower on narcissism and vulnerable subordinates). Moreover, given that vulnerable subordinates have less ability to voice their problems, support groups or a mentor could help them express their feelings. Additionally, specific training programs in self-esteem, self-confidence, and self-efficacy could help vulnerable individuals develop skills to cope with the narcissistic style of leadership and become more resilient [110]. Finally, empathy is proposed by Kohut as the main therapeutic factor to help accept own limitations and imperfections, decrease the grandiose fantasies or devaluation, and give meaning to one’s self.

## 5. Conclusions

We believed that, building on the biopsychosocial model, this study contributed to the growing body of evidence about the interaction between personal and situational factors in explaining workaholism and work engagement, a “bad” and a “good” form of heavy work investment, respectively. More specifically, our study showed that narcissism was positively associated with workaholism and its dimensions of WE and WC only in individuals facing a high workload. Furthermore, narcissism was positively associated with work engagement and its dimension of vigor and dedication (but no absorption) in employees with average levels of workload. Finally, workload exacerbated the relationship between narcissism and work engagement and its dimensions, so that these associations were stronger when the workload was high. Taken together, our results suggested that narcissism per se might give rise to work engagement, but not to workaholism. However, in a demanding work environment, narcissism might be associated with a less severe form of workaholism, characterized by high levels of both workaholism and work engagement (i.e., engaged workaholics) [78]. Accordingly, in terms of prevention and health promotion at work [111], interventions in organizational contexts should be aimed at dealing with both narcissism and workload, to prevent the onset of a more severe or clinical form of workaholism (i.e., disengaged workaholics) [78] in individuals with strong narcissistic components.

## Figures and Tables

**Figure 1 ijerph-17-04750-f001:**
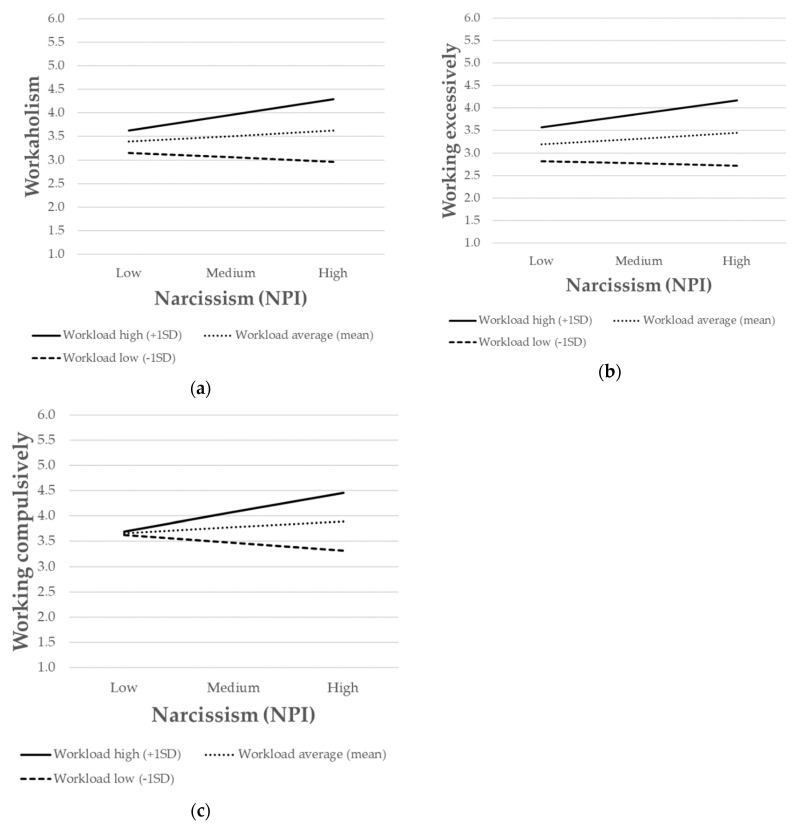
The interaction between narcissism and workload on (**a**) workaholism; (**b**) working excessively; (**c**) working compulsively. SD = standard deviation, NPI = narcissistic personality inventory.

**Figure 2 ijerph-17-04750-f002:**
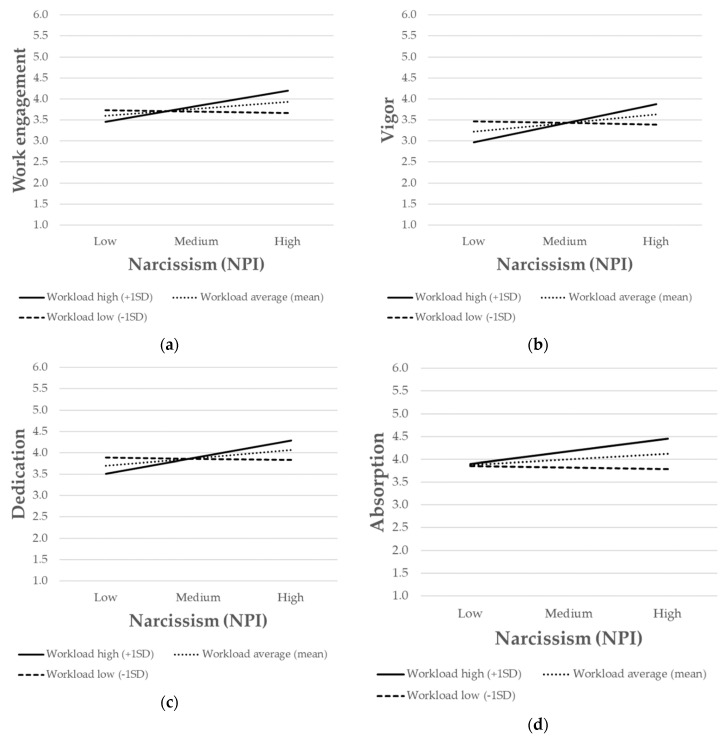
The interaction between narcissism and workload on: (**a**) work engagement; (**b**) vigor; (**c**) dedication; (**d**) absorption. SD = standard deviation.

**Table 1 ijerph-17-04750-t001:** Means, Standard Deviations, Cronbach’s Alphas, and Correlations between Study Variables (*N* = 217).

	*M*	*SD*	1	2	3	4	5	6	7	8	9
1. Narcissism ^1^	3.31	2.65	0.73								
2. Workaholism	3.52	1.04	0.15 *	0.87							
3. Working excessively	3.33	1.19	0.14 *	0.92 ***	0.83						
4. Working compulsively	3.79	1.17	0.12	0.81 ***	0.52 ***	0.87					
5. Work engagement	3.78	1.10	0.17 *	0.28 ***	0.20 **	0.30 ***	0.94				
6. Vigor	3.44	1.17	0.19 **	0.13	0.09	0.15 *	0.89 ***	0.87			
7. Dedication	3.89	1.35	0.15 *	0.20 **	0.16 *	0.21 **	0.93 ***	0.78 ***	0.94		
8. Absorption	4.01	1.15	0.12	0.43 ***	0.32 ***	0.46 ***	0.87 ***	0.64 ***	0.72 ***	0.85	
9. Workload	3.99	1.21	0.05	0.44 ***	0.47 ***	0.27 ***	0.07	0.01	0.03	0.16 *	0.79

Note: Internal consistency of the scales (Cronbach’s alpha) is displayed in the diagonal of the correlation matrix. ^1^ Items 6, 10, and 11 of the short version of the narcissistic personality inventory [44] behaved poorly in the confirmatory factor analysis and were not included in the composite scale score. * *p* < 0.05. ** *p* < 0.01. *** *p* < 0.001.

**Table 2 ijerph-17-04750-t002:** Results from Moderated Multiple Regression Analyses: Model 1, Model 2, and Model 3 (*N* = 217).

	Model 1—Workaholism		Model 2—Working Excessively		Model 3—Working Compulsively	
	***B***	***SE***	***B***	***SE***	***B***	***SE***
Narcissism	0.044	0.023	0.047	0.027	0.042	0.028
Workload	0.373 ***	0.051	0.458 ***	0.058	0.249 ***	0.062
Narcissism × workload	0.066 ***	0.019	0.054 *	0.022	0.083 ***	0.023
Total *R*^2^	0.257		0.259		0.137	
Change in *R*^2^	0.043		0.022		0.053	

Note: *B* = unstandardized regression coefficient; *R*^2^ = squared multiple correlation; *SE* = standard error. * *p* < 0.05. *** *p* < 0.001.

**Table 3 ijerph-17-04750-t003:** Results from Moderated Multiple Regression Analyses: Model 4, Model 5, Model 6, and Model 7 (*N* = 217).

	Model 4—Work Engagement		Model 5—Vigor		Model 6—Dedication		Model 7—Absorption	
	***B***	***SE***	***B***	***SE***	***B***	***SE***	***B***	***SE***
Narcissism	0.064 *	0.028	0.078 **	0.029	0.068 *	0.034	0.046	0.029
Workload	0.052	0.061	−0.004	0.064	0.016	0.075	0.145 *	0.064
Narcissism × workload	0.063 **	0.023	0.077 **	0.024	0.064 *	0.028	0.049 *	0.024
Total *R*^2^	0.066		0.081		0.046		0.057	
Change in *R*^2^	0.034		0.045		0.024		0.019	

Note: *B* = unstandardized regression coefficient; *R*^2^ = squared multiple correlation; *SE* = standard error. * *p* < 0.05. ** *p* < 0.01.

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
