# Peer review of "Is Narcissism Associated with Heavy Work Investment? The Moderating Role of Workload in the Relationship between Narcissism, Workaholism, and Work Engagement"

_ijerph, 2020, doi:10.3390/ijerph17134750_

Round 1

Reviewer 1 Report

This is a cross-sectional study that examined a moderating role of workload in the relationship between narcissism and two types of heavy work investments (i.e., workaholism and work engagement). The strong point of this article is to focus on the interplay between personal factor (narcissism) and work environment (workload). There are, however, a number of issues, as noted below, that need to be resolved and better addressed for the paper to make an even greater contribution to the field.

  1. (Introduction) Authors proposed hypotheses on the basis of the biopsychosocial model. However, this research did not deal with biological aspect. Please explain more about the rationale for building hypotheses on the basis of biopsychosocial model.
  2. (Introduction) The explanation of narcissism is redundant. Please condense it.
  3. (Data analysis) The findings can be confounded by demographic variables. Please also examine a model that includes demographic variables in Step 1 of the moderated multiple regressions analyses.
  4. (Conclusion) The conclusion part is redundant. Please refer only to ‘conclusion’ of this research.
  5. (References) The number of references (143) is too large. Please limit to the minimum required references.

Reviewer 2 Report

  1. 3 “At present, the definition of narcissistic personality has become widely used as that of “borderline.” Can you elaborate on what you mean? Borderline personality disorder? It’s unclear what is meant by borderline.
  2. Since much of your review of the literature for narcissism focuses on the clinical lens, it may be beneficial to clarify for the reader that your study is focusing instead on trait narcissism rather than a formal clinical diagnosis of NPD. It may also be worth considering adding a bit of literature review regarding trait narcissism in the organizational/work context since that is the focus of your paper.
  3. 9 – you mention that 3 items from the NPI-16 behaved poorly in your CFA and were thus removed. Please state which 3 items behaved poorly (either in text or in a footnote).
  4. Do you think there was any impact on the range of responses for narcissism on your findings, given that individuals could have scored a possible maximum of 13 on the NPI, but the average was only 3.31? Is this typical for the general population, and/or does it suggest range restriction?
  5. Did you consider testing for gender differences in these effects, since meta-analytic research suggests there are gender differences in narcissism? (Grijalva et al., 2015).

References:

Grijalva, E., Newman, D. A., Tay, L., Donnellan, M. B., Harms, P. D., Robins, R. W., & Yan, T. (2015). Gender differences in narcissism: A meta-analytic review. Psychological Bulletin141(2), 261-310.
